# Warm Mix Asphalt Binder Utilizing Water Foaming and Fluxing Using Bio-Derived Agent

**DOI:** 10.3390/ma15248873

**Published:** 2022-12-12

**Authors:** Marek Iwański, Anna Chomicz-Kowalska, Krzysztof Maciejewski, Mateusz M. Iwański, Piotr Radziszewski, Adam Liphardt, Jan B. Król, Michał Sarnowski, Karol J. Kowalski, Piotr Pokorski

**Affiliations:** 1Department of Transportation Engineering, Faculty of Civil Engineering and Architecture, Kielce University of Technology, 25-314 Kielce, Poland; 2Department of Building Engineering Technologies and Organization, Faculty of Civil Engineering and Architecture, Kielce University of Technology, 25-314 Kielce, Poland; 3Institute of Road and Bridges, Faculty of Civil Engineering, Warsaw University of Technology, 00-637 Warsaw, Poland

**Keywords:** warm mix asphalt, foamed bitumen, fluxing agent, highly modified asphalt binder, performance grade, DSR, BBR, MSCR

## Abstract

The present paper investigates the effects of simultaneous mechanical foaming using water and fluxing with a bio-derived agent on the properties of three distinct asphalt binders: 50/70 paving-grade bitumen, 45/80–55 polymer-modified bitumen, and 45/80–80 highly modified asphalt binder. The testing involved classical tests for assessing binder consistency (penetration at 25 °C, ring and ball softening point, Fraass breaking point, and dynamic viscosity) as well as performance tests (high and low Superpave critical temperatures and multiple stress creep recovery). The tests included assessment directly after asphalt binder foaming and were repeated after a 14-day period. It was shown that bitumen foaming had only short-term effects on the asphalt binders, which did not persist in the repeated tests after 14 days. The fluxing agent that was utilized caused significant changes in the consistency of all asphalt binders. The changes in the performance characteristics of the 50/70 and 45/80–55 binders were severe and amounted to a significant decrease in high-temperature performance of these binders. On the other hand, an improvement in all performance characteristics in the case of the 45/80–80 asphalt binder was observed as a result of the applied processes, particularly when measured 14 days after foaming. This study shows that the simultaneous use of foaming and the fluxing additive decreased the dynamic viscosity of the 45/80–80 binder, while improving its properties relating the pavement performance.

## 1. Introduction

New methods and techniques which would enable construction of long-lasting, durable, cleaner, and sustainable road pavements are currently being investigated. These efforts are motivated by many factors, most of which are economic, environmental, and health related.

Construction and maintenance of road infrastructure is critical for the world economies regardless of their development stage, but due to the high energy intensity required to process paving materials its impacts on the environment can be substantial and diverse. The production of asphalt mixtures involves heating of aggregates and asphalt binders to high temperatures in order to expel moisture from the aggregates and decrease the viscosity of the asphalt binder, enabling adequate mixing of the mixture’s constituents and obtaining a sufficient level of compaction during paving of the asphalt layer. Asphalt plants typically use burners to heat the aggregates, which typically make up approx. 94–97% of the asphalt mixture. These burners may utilize different types of fuels (gaseous, liquid, and even solid), and depending on the type of the fuel used, different types and amounts of the combustion products are emitted [1]. The products of high temperature combustion of fuels include but are not limited to carbon dioxide (CO_2_), carbon oxide (CO), sulfur oxides (SO_x_), nitrogen oxides (NO_x_), and dusts. These emissions are accompanied by the release of volatile organic substances from the hot asphalt binders in the form of volatile organic compounds (VOCs) and polycyclic aromatic hydrocarbons (PAHs) [2]. The mentioned compounds include greenhouse gasses, carcinogens, and other types of pollutants. It was found that the intensity of these emissions may be significantly reduced by decreasing the mixture’s production temperature [3,4], which also directly corresponds with the decrease in the energy demand of the production process and decrease in the occupational risks [5].

Due to the significant benefits of decreasing processing temperatures in asphalt paving, major efforts have recently been made to investigate the effects of different methods for producing warm mix asphalt and half-warm mix asphalt mixtures [6,7,8]. Some of the investigated avenues include the introduction of proprietary techniques and additives (e.g., liquid and solid warm mix additives) [9,10,11,12], asphalt binder foaming using zeolite and other water-bearing minerals [13,14,15,16], direct water injection [17,18,19,20], injection of alcohols [21,22], and the use of other kinds of foaming mediums, e.g., maleic anhydride [23].

In recent years, a range of new types of fluxing agents have been introduced, which enable the production of highly performing asphalt mixtures at lowered temperatures. Applying fluxing agents (additives) allows the lowering of bituminous binder viscosity and technological process temperature by 20–40 °C for asphalt mixture production and placement. This phenomenon provides ecological benefits, improves the working environment, and extends the construction season. In addition, the mixture hauling distance can be extended [24]. Fluxing agents have also been used in the production of asphalt mixtures using RAP (reclaimed asphalt pavement) material. They have the properties of refreshing the bitumen contained in the RAP and changing the stiffness of the binder and the asphalt mixture. They also allow the use of larger amounts of RAP without deteriorating the properties of the asphalt mixture [25]. Asphalt mixtures with a fluxing agent are also good as an anti-cracking component or in reinforcing layers on surfaces damaged by cracks. Such mixtures are characterized by increased fatigue life and resistance to atmospheric factors, such as low temperatures and water [26,27]. A fluxing agent can also be used in surface treatment technology as well as in the production of bitumen emulsions and for the production of asphalt mixtures intended for storage, where the use of a fluxing agent ensures adequate workability during its storage.

Examples of applications of plant-derived fluxing agents are known in the literature. One such agent is an additive called Bioflux, patented at the Warsaw University of Technology [28]. It is a product of the rapeseed methyl ester (RME), which is subjected to oxidation in the presence of a metal catalyst and organic peroxide. Organic acid salts of cobalt are used as a metal catalyst, and cumene hydroperoxide is used as a polymerization promoter. The concept of the production and mechanism of the oxypolymerization reaction has been described in detail in [29,30]. Bioflux, when added to bitumen, lowers its viscosity, which allows for lowering technological temperatures and producing an asphalt mixture in the WMA (warm mix asphalt) technology [28,30]. The work of Kowalski et al. [24] presents an analysis of the possible use of Bioflux as a refreshing binder agent in asphalt mixes with RAP, which enables the addition of RAP in larger amounts. The addition of Bioflux improves the homogeneity of the mixture, which affects the compaction process. The mixture’s fatigue life and low-temperature properties are also improved [27]. Another additive called Oleoflux consists of fatty acid methyl ester from sunflower oil. It lowers the bituminous binder’s viscosity, promotes the binder’s adhesion to the aggregate, and improves the water resistance of WMA asphalt mixtures [31]. Another example of a plant-derived fluidizing additive, the Green Seal additive, can be mentioned. It consists of a liquefied vegetable resin with monoalkyl esters from vegetable oils and animal fats. It is primarily an additive that improves binder adhesion [31]. An additive content of 0.5% has been found to have little effect on the binder properties, although the mixture blend homogeneity is improved. For a 5% additive, the binder becomes very soft even at room temperature—suggesting that it can be used as an additive in the recycling process to refresh the old binder. With a 5% content of fluidizers, the compressive strength and complex modulus drop significantly due to excessive softening of the binder [31]. Used cooking (post-frying) oils can also be used as liquefying additives. Waste Cooking Oil (WCO) is often added to asphalt mixtures in liquid form or modified bitumen before mixing with an aggregate to refresh the properties of the reclaimed binder in mixtures containing RAP. Therefore, they are used as rejuvenators [32]. Another plant-derived additive also used as a rejuvenator is the so-called W-oil, a by-product of fatty acid extraction from vegetable oil, obtained by distillation after the acidification of the vegetable oil. It is also used to improve the workability of the asphalt mixture [33]. Other additives are also known from raw plant materials that act as refreshing agents, such as an additive obtained from pine biomass as a result of pyrolysis [34]. Among the liquefiers of plant origin, one can also distinguish distilled tall oil, a by-product of paper production. Tall oils are used as emulsifiers, anti-stripping additives, additives to asphalt mixtures produced in the “warm” technology, and rejuvenators [34].

Because the incorporation of asphalt foaming requires only a one-time investment in the asphalt plant, it can be implemented at very small cost into the production practice in conjunction with other WMA techniques and additives to possibly improve the quality of the resulting mixtures further. Such possibilities have been studied regarding the simultaneous use of asphalt foaming and the use of different additives, including liquid and solid warm mix additives [12,35,36], often yielding favorable outcomes, e.g., substantially decreased binder viscosity or improved high-temperature performance.

In the recent years, much work has been conducted on improving the overall performance characteristics of warm mix asphalts to make them adequate for long-lasting durable pavements, even when subjected to high volumes of traffic. Some of these attempts include the use of polyphosphoric acid [37], crumb rubber [38,39,40], and polymer-modified bitumen [41] in formulating warm mix asphalts.

A recent study has investigated the possibility of utilizing the Bioflux additive to decrease the production temperatures of asphalt mixtures with highly (elastomer) modified asphalt binders [42]. In this study, significant decreases in dynamic viscosity of the modified binder were achieved, without sacrificing the exceptional high-temperature performance of the asphalt binder. A different study has also shown that these types of highly modified asphalt binders are well suited for being foamed using mechanical water foaming [43].

The aim of the present study was formulated based on the presented state of the art and on the need for innovative solutions allowing construction of highly performing, yet sustainable pavements. The paper investigates the effects of simultaneous water foaming and fluxing on classical and performance-related properties of three distinct asphalt binders used for surface course paving applications in Poland (paving-grade bitumen, polymer-modified bitumen, and a highly modified bitumen). The selection of the asphalt binders used in the study was based on their widespread use and their distinct rheological characteristics arising from different degrees and types of modification. The study involved testing of the asphalt binders directly after fluxing and foaming and 14 days after the process to evaluate the effects due to stabilization of fluxed foamed bitumen in physical and chemical processes.

## 2. Materials and Methods

### 2.1. Materials

#### 2.1.1. Asphalt Binder

The present study utilized three asphalt binders, 50/70 paving-grade bitumen, 45/80–55 polymer-modified bitumen, and 45/80–80 highly modified bitumen (HiMA), used in Poland [44] typically for producing asphalt surface courses intended for low, medium, and heavy trafficked roads, respectively. The specific asphalt binders were selected for their distinct functional characteristics despite their similar penetration ranges. The asphalt binders were commercially sourced from Orlen Asfalt (Poland). The basic characterization of these base asphalt binders is presented in Table 1.

Although the base asphalt binders were comparable in terms of their penetration at 25 °C, their remaining properties differed greatly due to the specific polymer modifications. The polymer-modified asphalt binders had greatly increased plasticity ranges defined as the difference between their softening points and Fraass breaking points. Additionally, the dynamic viscosities of the asphalt binders were significantly affected by the modification.

#### 2.1.2. Bio-Derived Fluxing Agent (Bioflux)

The fluxing agent was produced using pure fatty acid methyl esters derived from rapeseed oil (RME) without adding anti-ageing additives, based on the experience from previous studies [25,29]. RME, widely used as an additive to bio-fuels, contains anti-ageing additives that may block hardening and stiffness recovery in bitumen and asphalt mixtures. Selected RME properties used in the study are shown in Table 2.

The foaming process of bitumen can be improved by adding oils or oil esters which lower the viscosity of the binder. This makes the foaming process more efficient. Nevertheless, the binder liquefaction effect is not favorable due to the properties of asphalt mixtures because of lowering their stiffness. Therefore, it is reasonable to use liquefication additives which, during the production process of an asphalt mixture, reduce its consistency, and when used in the road structure, its original properties are slowly restored in the cross-linking process. It is possible through the oxypolymerization reaction and the availability of double unsaturated bonds in oils. Vegetable oils comprise a variable number of double bonds depending on the composition, and RME is needed to be activated because of its natural limits. Cross-linking efficiency depends on the number of double bonds and their position in the aliphatic chain of fatty acids. Thus, the pure RME was subjected to an oxidation reaction in the presence of oxidation promoters for activation:−Cobalt catalyst: 0.1% m/m converted to metal;−Polymerization initializer: cumene hydrogen peroxide 1.0% m/m.

The reaction was conducted in a laboratory reactor with a 0.3 ratio of reactor diameter to RME height and with an initial temperature of 25 °C. It was oxidized for two hours with an airflow value of 500 L/h per 1 kg of the product. The final product was kept in a sealed steel container at 5 °C until use.

### 2.2. Methods

#### 2.2.1. Design of Experiment

The experimental plan was set up to investigate the combined effects of foaming and the addition of the fluxing agent in different timeframes on the properties of three distinctly different asphalt binders. All tests were conducted in terms of the asphalt foaming and the time that elapsed after foaming:−Non-foamed asphalt binders (designation: NF);−Foamed asphalt binders, tested immediately after foaming (designation: F);−Foamed asphalt binders, tested 14 days after foaming (designation: F-14d).

The 14-day testing time was selected based on the experience from previous studies [29] as a time when rheological properties stabilize once foamed bitumen recovers its original properties and most of the flux cross-linking process ends. During the 14-day period, asphalt binders were kept at room temperature in sealed steel containers. The effects of the fluxing agent (Bioflux) were investigated in all mentioned cases using its three dosing rates, 1%, 2% and 3%, which were defined in preliminary testing (time-related stabilization and experiences from previous studies [25,29,30]). Additionally, non-fluxed binders were tested (0% Bioflux).

In relevant cases, the asphalt binders (after adding fluxing agent and asphalt foaming when applicable) were subjected to the rolling thin film oven test (RTFOT, Matest, Treviolo, Italy) and pressure ageing vessel (PAV, Prentex, Sunnyvale, TX, USA) ageing.

#### 2.2.2. Testing Methods

The investigated asphalt binders were tested according to normalized procedures for evaluating the following characteristics for their basic classification properties using automated apparatuses: penetration, softening point, and Fraass breaking point. The performance properties were assessed through a direct shear rheometer (TA Instruments DHR-2, New Castle, DE, USA) and a bending beam rheometer (Applied Test Systems, Butler, PA, USA). The testing methodologies included:−Classical properties of asphalt binders:
oPenetration at 25 °C (EN 1426);oSoftening point (EN 1427);oFraass breaking point (EN 12593);oElastic recovery (EN 13398)–only polymer-modified bitumen;oDynamic viscosity using a rotational viscometer at 90 °C, 110 °C and 135 °C (EN 13702-2).−Performance characteristics of asphalt binders:
oOscillatory dynamic shear testing for obtaining G*/sin(δ) values and calculating high critical temperatures (G*/sin(δ) = 1.0 kPa before RTFOT or G*/sin(δ) = 2.2 after RTFOT, EN 14770);oMultiple stress creep recovery testing using DSR for evaluating non-recoverable compliance and recovery after RTFOT (EN 16659);oBending beam rheometer testing for evaluating low critical temperatures after RTFOT + PAV (S_60_ = 300 MPa or m_60_ = 0.3, EN 14771).

The investigated binders’ fluxing was commenced before testing and foaming of asphalt binders as needed by the experimental plan. The metal containers filled with adequate binders were heated in laboratory ovens, and the fluxing agent was mixed with the binders mechanically (if it was not foamed) or directly in the laboratory foamer. The foaming of asphalt binders was conducted using a WLB-10S laboratory foamer (Wirtgen GmbH, Windhagen, Germany) with 2% of foaming water content as in [43]. The foaming air and water pressures were set to 500 and 600 kPa, respectively. The temperature of asphalt binders during foaming were set to 155 °C for all formulations based on the findings of the previous study [43].

The test results provided in the following sections are evaluated in terms of three main investigated factors using bar and line graphs:−Type of the asphalt binder (50/70, 45/80–55, and 45/80–80);−Asphalt foaming (NF, F, F-14d);−Bioflux additive content (0%, 1%, 2%, and 3%).

When the data provided in bar graphs represents means of multiple measured values, it is supplemented with 95% confidence intervals for the calculated means.

## 3. Results

### 3.1. Classical Properties of the Investigated Asphalt Binders

#### 3.1.1. Penetration at 25 °C, Softening Point, and Fraass Breaking Point

The evaluations of penetration at 25 °C, softening point, and Fraass breaking point were performed to measure the effects of foaming and fluxing on the classification properties of asphalt binders used in the countries using the CEN standardization framework.

Figure 1 presents the effects of foaming and the amount of Bioflux agent on the penetration at 25 °C of the investigated asphalt binders.

The evaluation of the effects of the fluxing agent on the values of penetration at 25°C showed that the increase in the amount of the additive significantly changed the consistency of all three asphalt binders, increasing their penetration. For most binders, the change in penetration was proportional to the amount of the Bioflux additive in its dosing range from 0% to 2%. The change in penetration caused by every percent of Bioflux content in non-foamed asphalt binders amounted to about 25 · 0.1 mm for the 50/70 and 45/80–55 binders and approx. 18 · 0.1 mm for the 45/80–80 highly modified binder. A further increase in the additive content from 2% to 3% resulted in a disproportionate increase in penetration by 56 · 0.1 mm in the case of 50/70 bitumen, and by 32 · 0.1 mm and 28 · 0.1 mm for the 45/80–55 and 45/80–80 asphalt binders, respectively. Compared to the non-fluxed asphalt binders, the 3% addition of Bioflux agent resulted in total increases of 105 · 0.1 mm, 82 · 0.1 mm, and 63 · 0.1 mm in the 50/70, 45/80–55, and 45/80–80 asphalt binders, respectively, which amounted to 170%, 115%, and 80% changes.

When measured after binder foaming, these effects of utilizing the fluxing agent were similar in magnitude. The differences in the foamed asphalt binders amounted to a ± 5% range in the non-foamed binders, but in general, the foamed asphalt binders were characterized by slightly decreased penetration. This decrease was most prominently observed in the 45/80–80 asphalt binder.

Based on the evaluation of the softening point test results presented in Figure 2, it can be concluded that in the case of all bituminous binders, the addition of Bioflux caused a noticeable, gradual decrease in their softening points, along with an increase in the content of the additive in the binders.

For the non-foamed binder 50/70 bitumen, increasing the content of Bioflux additive by 1% in the entire dosing range from 0% to 3% decreased the softening point by an average of 2.8 °C. The softening point of the polymer-modified bitumen 45/80–55 decreased by an average of 2.5 °C, and the highly modified 45/80–80 binder by 3.3 °C.

Directly after foaming, most of the investigated binders were characterized by a softening point lower by 1 °C than non-foamed binders, and in the tests after 14 days, this difference was no longer observed. In cases of the highly modified 45/80–80 binder, the softening point was reduced to a significantly higher degree immediately after and 14 days after foaming when dosed with 1% and 3% of Bioflux agent.

The results of the Fraass breaking point presented in Figure 3 show that the combined use of foaming and the addition of the fluxing agent had measurable impacts on the values of this parameter, which was specifically evident in the case of the 50/70 and 45/80–80 asphalt binders. The polymer-modified 45/80–55 asphalt binder had a similar performance in this scope, regardless of its foaming.

Foaming alone caused increases in the breaking point temperatures, which amounted to 1 °C in all three asphalt binders. Typically, this increase was seen both directly after foaming and after 14 days, with the 45/80–80 binder being the exception—the result was the same as in the non-foamed asphalt binder.

The Bioflux agent had a far greater impact on this characteristic. In most of the tested binders, adding the fluxing agent decreased the breaking point temperatures, both in non-foamed and foamed binders. The magnitude of these changes varied depending on the type of the base binder and the foaming state. The greatest changes were in the non-foamed 50/70 paving-grade bitumen, characterized by breaking point temperatures typically observed for polymer-modified binders. As in the foamed non-fluxed asphalt binders, the fluxed binders directly after foaming exhibited increased breaking point temperatures, which decreased again after 14 days.

The investigated processes had different effects on the evaluated classical properties of the asphalt binders. The major impacts on the consistency could be attributed to the addition of the fluxing agent. Foaming alone had only small effects on these properties, and they were dependent on the type of the asphalt binder, as also found by Martinez-Arguelles et al. [45].

The results of statistical analyses presented in Table 3 show that the Bioflux content significantly affected all evaluated classical properties of the tested binders. It is also shown that there was no evidence that the foaming significantly changed the values of penetration at 25 °C. On the contrary, the effects of both factors were statistically significant in the case of the softening point in all tested asphalt binders. The softening point values in the F-14d samples were not affected by the process. The Fraass breaking point was affected by the foaming process only in the case of 50/70 and 45/80–55 asphalt binders.

#### 3.1.2. Elastic Recovery

The results of an evaluation of elastic recovery of 45/80–55 and 45/80–80 asphalt binders after RTFOT short-term aging are presented in Figure 4.

Based on the presented results, it can be stated that both evaluated polymer-modified asphalt binders were characterized by very high values of elastic recovery (>80% and >90% for the 45/80–55 and 45/80–80 binders, respectively), which were not significantly affected by the process of foaming or fluxing. Asphalt foaming alone had only small effects on the elastic recovery directly after the process. After 14 days, the R_E_ values were indistinguishable from those in non-foamed 45/80–55 and 45/80–80 binders. The addition of the Bioflux fluxing agent had only minor effects on this characteristic in the case of both binders. The 45/80–55 asphalt binder was most prominent after being measured 14 days after foaming—the greatest change amounted to a 1.7 percent point increase in elastic recovery with the 3% addition of Bioflux. The fluxing agent had larger effects on the elastic recovery of the highly modified asphalt binder, however, they could still be objectively regarded as minor in magnitude, amounting to a 4.2 percent point increase in elastic recovery in the non-foamed binder, 2.0 percent point increase directly after foaming, and 2.8 percent point increase 14 days after foaming. In only one case—directly after foaming the blend with 1% of Bioflux—a 3.6 percent point decrease in this characteristic was observed.

The results of statistical analyses presented in Table 4 show that the applied statistical models explain only a very small fraction of the observed variability in the data (low adj. R^2^ values). Only in the case of the 45/80–55 asphalt binder the applied model was marginally applicable and showed statistical significance of both the effects of Bioflux content and foaming in all instances. It should be stated again that the observed differences in measured recovery values were marginal. In the case of the 45/80–80 asphalt binder, the statistical model was inadequate to characterize the observed variability, however, applying a more complex model would lead to overfitting and would not add more insight to the data.

#### 3.1.3. Dynamic Viscosity

The dynamic viscosity of an asphalt binder and how it changes depending on the temperature are among the most important factors determining technological (production, laying, and compaction) temperatures of asphalt mixtures produced by traditional (hot) methods. Among the different methods for evaluating the flow properties of asphalt binders, the methods utilizing viscometers and shear rheometers are especially valuable given their capacity to control the temperature and shear rate. The dynamic viscosity tests were carried out using the coaxial cylinder method at temperatures of 90 °C, 110 °C, and 135 °C (within the assumed technological temperatures for the production and compaction of future mineral–asphalt composites) for all three types of reference binders without and with the addition of the Bioflux fluxing agent, before foaming (NF), immediately after foaming (F), and after 14 days from foaming (F-14d). The results of the dynamic viscosity test as a function of temperature are presented in Figure 5. The changes in the values of dynamic viscosity due to fluxing and foaming are additionally presented in Figure 6 in relation to the base asphalt binders.

The presented results indicate that adding Bioflux and foaming significantly reduced the dynamic viscosities of the investigated asphalt binders.

The polymer-modified asphalt binders were characterized by higher viscosities in the whole range of testing temperatures than the 50/70 paving-grade bitumen. Specifically, the dynamic viscosity of the highly modified bitumen was approx. 10× higher than the other binders. At the same time, this 40/80–80 binder experienced the greatest decrease in dynamic viscosity due to the increase in the testing temperature. Regarding the effects of foaming, it can be seen that in most cases, the reduction in dynamic viscosity was reversed and nonsignificant 14 days after foaming. Additionally, these effects were more pronounced in polymer-modified asphalt binders than in the paving-grade bitumen. In single cases, it was observed that foaming and the fluxing agent caused the dynamic viscosity to increase, which could be caused by the interference of the foaming effects. In most cases, the effects of foaming and fluxing agent did not compound, and the reduction in viscosity in fluxed foamed binders was very similar to that seen in non-foamed asphalt binders. Exceptions from this were seen when 3% of the fluxing agent was used. In general, the increase in the dosing of fluxing agent resulted in a decrease in the dynamic viscosity.

The changes in dynamic viscosity were greatest and most consistent throughout the testing temperatures in the 45/80–80 highly modified asphalt binder. The reductions in dynamic viscosity amounted to approximately 44% in the 50/70 binder, 84% in the 45/80–55 binder, and 94% in the 45/80–80 binder.

### 3.2. Performance Characteristics of the Investigated Asphalt Binders

#### 3.2.1. High and Low Critical Temperatures Measured in Dynamic Shear and Bending Beam Rheometers

The high and low critical temperatures were calculated based on the measurements in dynamic shear and bending beam rheometers. The high critical temperatures were evaluated based on the conditions for the high temperature stiffness (G*/sinδ), while the low critical temperatures were based on the creep stiffness (S60) conditions and the m-value. Figure 7 shows the spans between the investigated binders’ low and high critical temperatures.

Foaming of the asphalt binders had only short-term effects on their critical temperatures. It was found that directly after foaming, the high critical temperatures of the binders 50/70, 45/80–55, and 45/80–80 decreased by 1.2 °C, 1.2 °C, and 0.4 °C, respectively. These changes were mostly reverted after 14 days. Similarly, asphalt foaming decreased the low critical temperatures, and their effects were partially reversed with time. These effects resulted in an overall increase in the span between low and high critical temperatures of the 50/70 and 45/80–80 binders, and almost no change in this characteristic in the case of the 45/80–55 polymer-modified asphalt binder.

The addition of the fluxing agent decreased both high and low critical temperatures in the 50/70 and 45/80–55 asphalt binders. In the 50/70 paving-grade bitumen, this decreased the span between the low and high critical temperatures, but in the case of the 45/80–55 asphalt binder, these ranges were increased by a small amount (<2 °C). In the case of the highly modified 45/80–80 asphalt binder, these effects were mixed, but in general, unlike other binders, both low- and high-temperature performance was improved when 1% and 3% of Bioflux was used.

The results of statistical analyses presented in Table 5 show that in regard to the 50/70 and 45/80–55 asphalt binders the applied statistical models are adequate and explain most of the observed variability (adj. R^2^ values higher than 0.9). The high critical temperature of the 50/70 asphalt binder was significantly affected only by the Bioflux content, while for the 45/80–55 asphalt binder the effects of foaming were also significant directly after the process. In the case of the 45/80–80 asphalt binder, the applied statistical model explained only a very small portion of the variability in the data, which was influenced in a more complex, nonlinear manner, requiring additional experiments. The low-temperature performance of all three asphalt binders was significantly affected by the Bioflux content and foaming process.

#### 3.2.2. Multiple Stress Creep Recovery Performance

Figure 8 shows the multiple stress creep recovery performance evaluated at 3.2 kPa shear stress and a temperature of 60 °C. As for the general assessment of the effect size, the greatest effect on this characteristic could be attributed to the presence and severity of modification with the fluxing agent. It should be noted that while in the 50/70 and 45/80–55 binders increased concentration of Bioflux resulted in increased non-recoverable compliance (J_nr 3.2 kPa_), this was not the case with the highly modified 45/80–80 asphalt binder, specifically after foaming. Foaming alone had only minor effects and mostly caused only slight changes in this parameter.

The effects of the investigated processes on the non-recoverable compliance of the 50/70 and 45/80–55 asphalt binders were consistent and predictable, given the aforementioned effects on dynamic viscosity and other characteristics. The addition of fluxing agent significantly increased the J_nr 3.2 kPa_ values, and foaming slightly contributed to these increases. On the other hand, a strong interaction was observed between foaming and the presence of the fluxing agent in the 45/80–80 highly modified asphalt binder, which caused substantial decreases in non-recoverable compliance, both directly after foaming and after 14 days. These results were in line with the high-temperature performance of these binders under oscillatory loading (high critical temperature).

Figure 9 presents the results of MSCR recovery tests. The 50/70 asphalt binder was omitted in this figure as it exhibited no recovery in the MSCR test conducted at 60 °C, a result typical for asphalt binder without an elastomeric modification. On the other hand, the 45/80–55 and 45/80–80 polymer-modified asphalt binders recorded high values of this parameter.

Foaming alone had only very small effects on the values of the recovery of the asphalt binders evaluated in the MSCR test. In the case of the foamed 45/80–55 polymer-modified binder, 1 and 3 percent point decreases were observed directly after foaming and after 14 days, respectively, while the highly modified binder saw only a 1 percent point decrease in this parameter. On the other hand, adding the fluxing agent caused a significant decrease in the recovery of the 45/80–55 asphalt binder. A decrease from 59% to as low as 44% was observed when a 3% Bioflux blend was tested 14 days after foaming. The 45/80–80 asphalt binder remained mostly unchanged in this scope, regardless of the amount of fluxing agent used.

The results of statistical analyses presented in Table 6 show that the MSCR performance of the 50/70 and 45/80–55 asphalt binders could be explained by the applied statistical models to a high degree. The performance of the 45/80–80 asphalt binders has, however, exhibited irregularities, which particularly after foaming could not be easily encompassed by the linear statistical models. Only the effects of the Bioflux additive were proven to be statistically significant in the case of the MSCR performance of the 50/70 and 45/80–55 binders.

## 4. Multivariate Optimization

A multivariate optimization approach was utilized to objectively quantify the various measured characteristics of investigated asphalt binders and rank them according to their performance. The method used desirability functions and a desirability index to evaluate multiple variables simultaneously, normalize their responses, and aggregate them to a single numerical value. The utilized desirability approach was extensively described in [46].

The analysis included step-linear-step desirability functions, defined by lower and upper specification limits (LSL and USL, respectively), which normalized all evaluated binder characteristics to the 0–1 range. In this framework, binder characteristics below the lower specification limit (the least desired value) are assigned desirability values of 0, while the values above the upper specification limit are assigned desirability values of 1. Intermediate values are interpolated linearly. The resulting set of desirability values (referred to as partial desirabilities) characterizing, e.g., different performance characteristics of a material, are aggregated using a geometric mean, the value of which is called the desirability index (DI).

To evaluate the performance of the investigated binders, high critical temperature, low critical temperature, non-recoverable compliance, and MSCR recovery were used in the multivariate optimization approach. The natural characteristic of the desirability index, defined as mentioned above, is that it amounts to 0 whenever any of the characteristics of the evaluated case fail to meet the lower specification limit. In this analysis, the aim was to obtain meaningful information regarding all of the experiments. Therefore, the low specification limits were set below the minimums in the data. The LSL values were set to 99% of the minimum observed values in the respective characteristics, and because the minimum observed MSCR recovery values were equal to 0, the LSL was set to −1 in this case. The specific values of the specification limits in the analysis are presented in Table 7.

The values of the computed partial desirabilities for all investigated asphalt binders are presented in Figure 10. The computed values reflect the investigated binders’ actual characteristics, but applying the desirability functions normalized the responses to the (0;1) range. Because the specification limits were common for all binders, direct comparisons between different asphalt binders and treatments can be made. The very low, although non-zero, values of partial desirability of the 50/70 asphalt binder in the scope of its MSCR recovery are the effect of the specifications set out in Table 7 and the negligible test results.

Based on the values of partial desirabilities, the desirability index for each type of the investigated asphalt binder was calculated and is presented in Figure 11. The desirability values of different groups of binders (based on the 50/70, 45/80–55, and 45/80–80 asphalt binders) varied greatly due to their different performance characteristics. The overall lowest DI values were recorded for the 50/70-based asphalt binders, which were in the range of 0.04–0.25. This was attributed mainly to the extremely poor recovery characteristics, although the remaining properties were also clearly less desirable than in the polymer-modified asphalt binders. The 45/80–55 asphalt binders recorded intermediate DI values ranging from 0.52 to 0.63, and the 45/80–80 highly modified bitumen scored the highest DI values (0.73–0.96).

Regarding the evaluated performance characteristics, the most important effects of foaming and fluxing relate to the long-term case, represented in the non-foamed binders and at the end of the 14 days after foaming. In these cases, the mentioned processes resulted in an increased overall performance of all evaluated asphalt binders, regardless of the amount of the fluxing agent used. In the case of the 50/70 binder, the 2% addition of Bioflux was most favorable. With the 45/80–55 binder, the desirability indices were highest when 1% of the additive was introduced, while the highly modified 45/80–80 binder performed best when mixed with 1% and 3% of the fluxing agent.

## 5. Conclusions

The present study investigated the effects of simultaneous use of a bio-derived fluxing agent based on oxidized rapeseed oil methyl esters (Bioflux) and foaming using water on the properties of three distinct asphalt binders: 50/70 paving-grade bitumen, 45/80–55 polymer-modified bitumen, and 45/80–80 highly modified asphalt binder. The asphalt binders used in the study were selected due to their wide use in producing wearing courses and their similar consistency measured by penetration at 25 °C.

The study has shown distinct differences in the effects of the investigated processes of fluxing and foaming on the properties of asphalt binders, depending on the type of the binder and the properties in question:−The typical classification properties of the investigated binders (penetration, softening point, and Fraass breaking point) were very strongly influenced by the addition of the fluxing agent, and to a lesser degree by foaming; the penetration of the binders significantly increased, while the softening and breaking points decreased due to the addition of the fluxing agent.−The elastic recovery of the polymer-modified binders increase slightly with the addition of Bioflux (particularly the 45/80–80 asphalt binder), while foaming had negligible effects in this case.−Both foaming and fluxing resulted in a decrease in dynamic viscosity of all evaluated binders, but the effects were not additive, which resulted in that the foaming had little effect on the dynamic viscosity of asphalt binders when the Bioflux agent was added; the greatest relative reductions in dynamic viscosity were seen in the 45/80–80 asphalt binder, followed by the 45/80–55 binder.−When present, the effects of foaming were significant only directly after the process and did not persist when evaluated 14 days after foaming.

Significant results were observed in the evaluation of the performance properties of the asphalt binders:−Major deterioration of the high-temperature performance due to introduction of the fluxing agent in the 50/70 and 45/80–55 asphalt binders was observed, both before and after foaming,−The 45/80–80 highly modified bitumen recorded decreased high critical temperatures and increased non-recoverable compliance in the non-foamed state, but after foaming, the fluxed 45/80–80 asphalt binder recorded improved high-temperature performance when 1% and 3% of Bioflux was used, which resulted in increased service temperature range, spanning up to 106.6 °C.

The response of the 45/80–80 highly modified asphalt binder to simultaneous fluxing and foaming was unlike the other tested bitumen, in which the application of the fluxing agent deteriorated the performance characteristics. The analyses have shown that simultaneous utilization of a Bioflux fluxing agent and water foaming resulted in significant improvements of the high- and low-temperature properties of the 45/80–80 highly modified asphalt binder, while decreasing its dynamic viscosity in the paving temperature range, potentially facilitating easier mixing and compaction of asphalt mixture.

## Figures and Tables

**Figure 1 materials-15-08873-f001:**
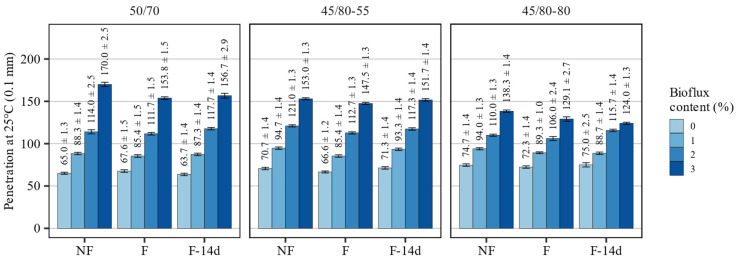
Results of penetration at 25 °C of the investigated asphalt binders.

**Figure 2 materials-15-08873-f002:**
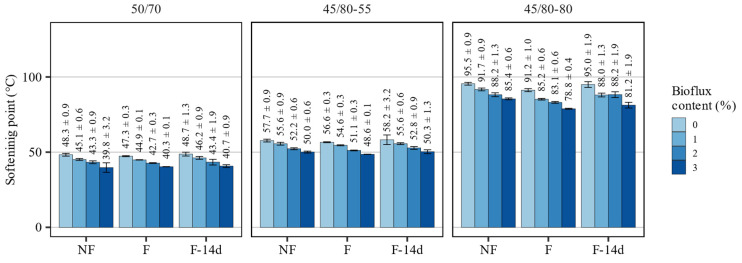
Results of softening point measurements of the investigated asphalt binders.

**Figure 3 materials-15-08873-f003:**
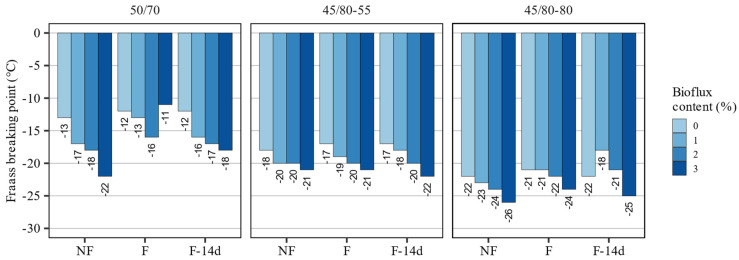
Results of Fraass breaking point measurements of the investigated asphalt binders.

**Figure 4 materials-15-08873-f004:**
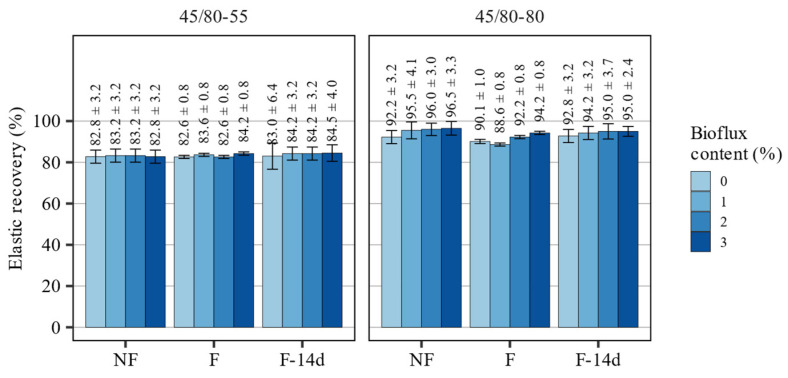
Results of elastic recovery measurements of the investigated asphalt binders.

**Figure 5 materials-15-08873-f005:**
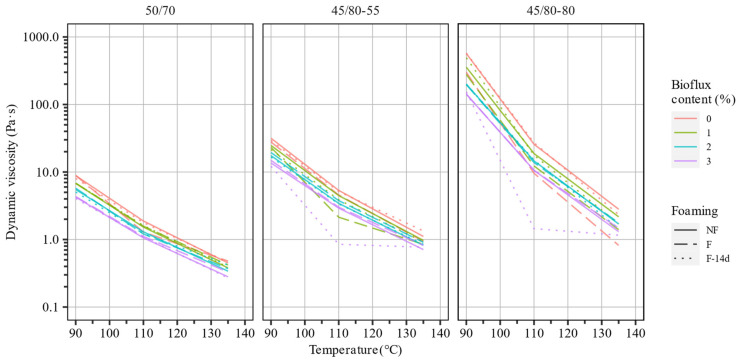
Results of dynamic viscosity measurements of the investigated asphalt binders.

**Figure 6 materials-15-08873-f006:**
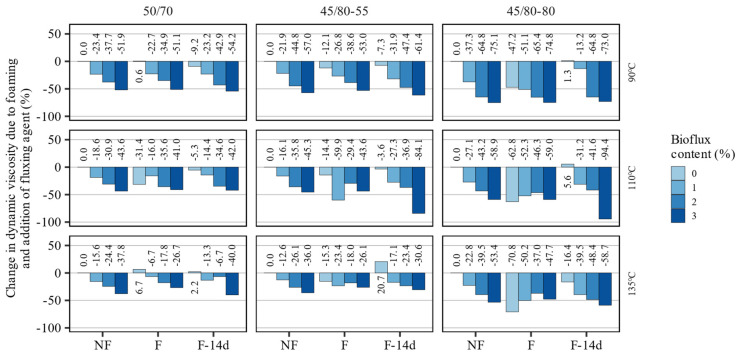
Relative changes in dynamic viscosity measurements of the investigated asphalt binders due to foaming and application of the fluxing agent.

**Figure 7 materials-15-08873-f007:**
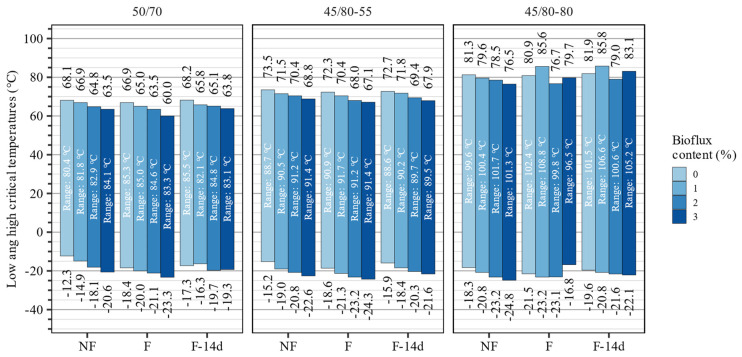
Low and high critical temperatures of the investigated binders assessed based on the dynamic shear and bending beam rheometer testing (based on G*/sin(δ) = 2.2 kPa after RTFOT and S(60) = 300 MPa or m(60) = 0.3 after RTFOT and PAV).

**Figure 8 materials-15-08873-f008:**
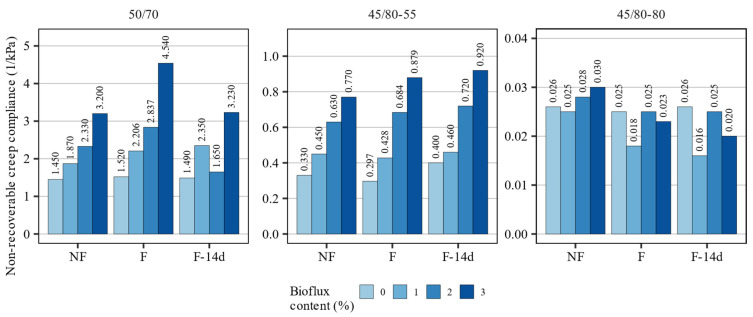
Non-recoverable creep compliance of the investigated binders measured at 3.2 kPa stress level and 60 °C.

**Figure 9 materials-15-08873-f009:**
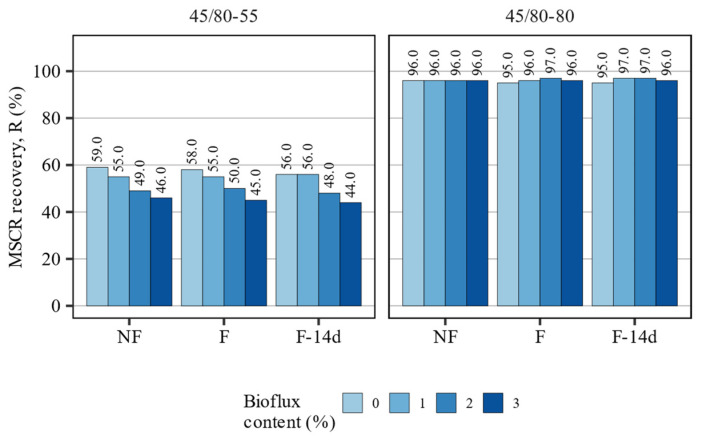
MSCR recovery R at 3.2 kPa stress level and 60 °C.

**Figure 10 materials-15-08873-f010:**
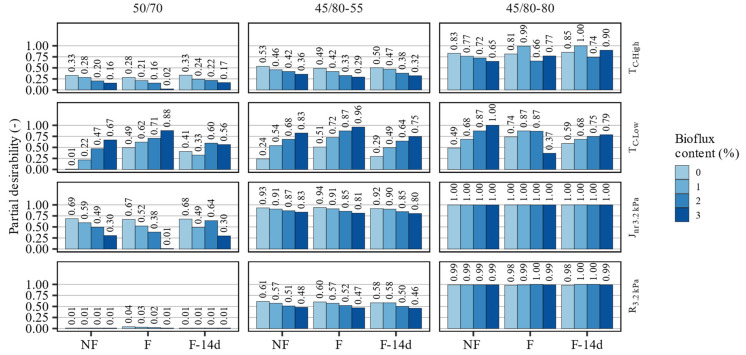
The computed partial desirability values with respect to the evaluated characteristics of investigated asphalt binders.

**Figure 11 materials-15-08873-f011:**
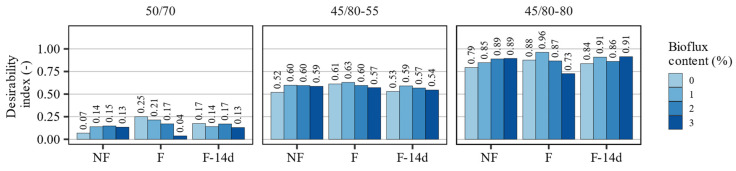
The desirability indices calculated using the multivariate optimization methodology.

**Table 1 materials-15-08873-t001:** Properties of the base asphalt binders used in the study.

Property	Unit of Measurement	Base Bitumen			Testing Method
		50/70	45/80–55	45/80–80	
Penetration at 25 °C	0.1 mm	65	71	75	EN 1426
Softening point	°C	48.2	57.8	95.5	EN 1427
Fraass breaking point	°C	−13	−18	−22	EN 12593
Dynamic viscosity at 135 °C	Pa·s	0.45	1.11	2.81	EN 13702–2
Dynamic viscosity at 135 °C after RTFOT	Pa·s	0.65	1.51	3.77	EN 13702–2

**Table 2 materials-15-08873-t002:** Selected properties of the pure RME for producing the Bioflux fluxing agent.

Property	Rapeseed Methyl Esters, RME
Iodine number, g I_2_/100 g	≥100
Viscosity at 25 °C, Pa·s	≤0.008
Acid number, mg KOH/g	≤0.50
Flashpoint, °C	≥180

**Table 3 materials-15-08873-t003:** The *p*-values calculated in analyses of variance for assessing the significance of the effects of Bioflux and foaming on the classical parameters of asphalt binders.

Asphalt Binder	Penetration at 25 °C			Softening Point (R and B)			Fraass Breaking Point		
*p*-Values	50/70	45/80–55	45/80–80	50/70	45/80–55	45/80–80	50/70	45/80–55	45/80–80
Intercept	<0.001	<0.001	<0.001	<0.001	<0.001	<0.001	<0.001	<0.001	<0.001
Bioflux	<0.001	<0.001	<0.001	<0.001	<0.001	0.002	0.012	0.005	0.154
Foaming: F	0.750	0.284	0.691	0.035	0.040	0.036	0.890	0.093	0.561
F-14d	0.882	0.976	0.764	0.193	0.328	0.585	0.854	0.021	0.358
Bioflux × Foaming: F	0.378	0.908	0.469	0.061	0.571	0.572	0.045	0.223	0.799
F-14d	0.612	0.780	0.253	0.741	0.852	0.475	0.449	0.035	0.932
Adj. R^2^	0.938	0.980	0.966	0.991	0.990	0.917	0.710	0.918	0.312

**Table 4 materials-15-08873-t004:** The *p*-values calculated in analyses of variance for assessing the significance of the effects of Bioflux and foaming on the elastic recovery of asphalt binders.

Asphalt Binder	Elastic Recovery	
*p*-Values	45/80–55	45/80–80
Intercept	<0.001	<0.001
Bioflux	0.011	0.196
Foaming: F	0.074	0.375
F-14d	0.030	0.985
Bioflux × Foaming: F	0.078	0.915
F-14d	0.124	0.782
Adj. R^2^	0.219	0.079

**Table 5 materials-15-08873-t005:** The *p*-values calculated in analyses of variance for assessing the significance of the effects of Bioflux and foaming on the high and low critical temperatures of asphalt binders.

Asphalt Binder	High Critical Temperature			Low Critical Temperature		
*p*-Values	50/70	45/80–55	45/80–80	50/70	45/80–55	45/80–80
Intercept	<0.001	<0.001	<0.001	<0.001	<0.001	<0.001
Bioflux	<0.001	0.000	0.303	<0.001	<0.001	0.028
Foaming: F	0.147	0.047	0.733	<0.001	0.004	0.054
F-14d	0.542	0.477	0.670	0.003	0.596	0.544
Bioflux × Foaming F	0.105	0.312	0.882	0.049	0.239	0.015
F-14d	0.567	0.551	0.550	0.009	0.239	0.252
Adj. R^2^	0.948	0.963	0.027	0.928	0.952	0.434

**Table 6 materials-15-08873-t006:** The *p*-values calculated in analyses of variance for assessing the significance of the effects of Bioflux and foaming on the non-recoverable compliance and MSCR recovery of asphalt binders.

Asphalt Binder	J_nr 3.2 kPa_			R_3.2 kPa_		
*p*-Values	50/70	45/80–55	45/80–80	50/70	45/80–55	45/80–80
Intercept	0.012	<0.001	<0.001	-	<0.001	<0.001
Bioflux	0.031	<0.001	0.435	-	<0.001	1.000
Foaming: F	0.951	0.389	0.631	-	0.831	0.531
F-14d	0.795	0.561	0.703	-	0.466	0.832
Bioflux × Foaming: F	0.216	0.121	0.601	-	0.929	0.439
F-14d	0.694	0.294	0.381	-	0.926	0.557
Adj. R^2^	0.759	0.957	0.018	-	0.918	0.228

**Table 7 materials-15-08873-t007:** The specification values of desirability functions used in the study.

Response Characteristic	Unit of Measurement	Data		Specification Limits	
		Minimum	Maximum	LSL	USL
High critical temperature (T_C-High_)	°C	60	85.8	59.4	85.8
Low critical temperature (T_C-Low_)	°C	−12.3	−24.8	−12.177	−24.8
Non-recoverable compliance (J_nr 3.2 kPa_)	1/kPa	4.54	0.016	4.5854	0.016
MSCR recovery (R_3.2 kPa_)	%	0	97	−1	97
Assigned desirability	-	-	-	0.0	1.0

## Data Availability

Data available on request from the corresponding author.

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
