# Peer review of "Warm Mix Asphalt Binder Utilizing Water Foaming and Fluxing Using Bio-Derived Agent"

_materials, 2022, doi:10.3390/ma15248873_

Round 1
Reviewer 1 Report
Interesting paper. However, the intro does not give enough information to support the need for this study. It does not seems to have much new information here. The use of flux, as mentionned in the paper, is more common now and so is foaming. The authors need to explain what makes this study different.
Good range of bitumen tests. The authors need to state better if the results are expected or not and if they fit with the literature. I do not see any unexpected results, so the authors need to work on the analysis. Also, why only short term aging was used and not long term like PAV, or extended PAV? It could have shown if the bioflux is stable with time. We have no information on the 14 days aging methodology. I guess PAV was used before BBR, but it's not mentionned
It would be interesting to have information about the foaming itself. what temperature was used? the same for all the mixes? That could significantly change the results.
here are specific comments:
- Table 1, why those specific binders were selected?
- why 14 days of aging?, why not longer?
- on most figures, we see error bars, but we do not know what they represent
-fig 6. how do you explain the few increases in viscosity?
-fig 7 is misleading. I would change the scale to highlight the difference between the results. Please explain how the gain from foaming is gone after the 14d aging; support your answer with the literature.
Author Response
Dear Reviewer,
We would like to voice our gratitude for the time and effort spent revising our paper titled “Warm mix asphalt binder utilizing water-foaming and fluxing using bio-derived agent”. We truly feel that your remarks are a significant contribution to the overall quality of our paper and enabled us to rectify its shortcomings.
After a thorough revision, we present you the corrected version of the manuscript for its assessment. Please find the detailed responses to your comments below.
Best regards,
Authors
Reviewer #1:
Interesting paper. However, the intro does not give enough information to support the need for this study. It does not seems to have much new information here. The use of flux, as mentionned in the paper, is more common now and so is foaming. The authors need to explain what makes this study different.
Thank you for the accurate remark. We have included the explanation in the last paragraph of the introduction:
“The selection of the asphalt binders used in the study was based on their widespread use and their distinct rheological characteristics arising from different degrees and types of modification. The study involved testing of the asphalt binders directly after fluxing and foaming and 14 days after the process, to evaluate the effects due to stabilization of fluxed foamed bitumen in physical and chemical processes.”
Good range of bitumen tests. The authors need to state better if the results are expected or not and if they fit with the literature. I do not see any unexpected results, so the authors need to work on the analysis. Also, why only short term aging was used and not long term like PAV, or extended PAV? It could have shown if the bioflux is stable with time. We have no information on the 14 days aging methodology. I guess PAV was used before BBR, but it's not mentioned
Thank you for your remark. We believe that the unexpected results covered by the paper include the increase in performance characteristics of the fluxed and foamed 45/80-80 asphalt binder, which was highlighted in the conclusions:
“The response of the 45/80-80 highly modified asphalt binder to simultaneous fluxing and foaming was unlike the other tested bitumen, in which the application of the fluxing agent deteriorated the performance characteristics. The analyses have shown that simultaneous utilization of Bioflux fluxing agent and water-foaming resulted in significant improvements of the high and low temperature properties of the 45/80-80 highly modified asphalt binder, while decreasing its dynamic viscosity in the paving temperature range, potentially facilitating easier mixing and compaction of asphalt mixture.”
In section 2.2.1 we have added information regarding the handling of asphalt binders during the 14 day period:
“During the 14 day period asphalt binders were kept in room temperature in sealed steel containers.”
We have supplemented section 2.2.2 with the information that the presented MSCR results were obtained on RTFOT aged samples and the BBR samples were subjected to RTFOT+PAV ageing
It would be interesting to have information about the foaming itself. what temperature was used? the same for all the mixes? That could significantly change the results.
Thank you for the accurate remark. Foaming conditions were the same for all mixtures. Adequate information was added to section 2.2.2:
„The foaming air and water pressures were set to 500 and 600 kPa respectively. The temperature of asphalt binders during foaming were set to 155°C for all formulations based on the findings of the previous study [43].”
here are specific comments:
- Table 1, why those specific binders were selected?
Thank you for the thoughtful remark. We have supplemented the last paragraph of the introduction:
“The aim of the present study was formulated based on the presented state of the art and on the needs for innovative solutions allowing construction of highly performing, yet sustainable pavements. The paper investigates the effects of simultaneous water-foaming and fluxing on classical and performance related properties of three distinct asphalt binders used for surface course paving applications in Poland (paving grade bitumen, polymer modified bitumen and a highly modified bitumen). The selection of the asphalt binders used in the study was based on their widespread use and their distinct rheological characteristics arising from different degrees and types of modification. The study in-volved testing of the asphalt binders directly after fluxing and foaming and 14 days after the process, to evaluate the effects due to stabilization of fluxed foamed bitumen in physi-cal and chemical processes.”
and the first paragraph of section 2.1.1 (L147-150):
„(…) used in Poland [44] typically for producing asphalt surface courses intended for low, medium and heavy trafficked roads, respectively. The specific asphalt binders were selected for their distinct functional characteristics despite similar penetration range.”
- why 14 days of aging?, why not longer?
Thank you for the remark. The 14 days testing time was selected based on the experience from previous studies [Gawel, I.; Pilat, J.; Radziszewski, P.; Niczke, L.; Krol, J.; Sarnowski, M. Bitumen fluxes of vegetable origin. Polimery/Polymers 2010, 55, 55–60, doi:10.14314/polimery.2010.055] as a time when rheological properties stabilize once foamed bitumen recovers its original properties and most of the flux crosslinking process ends.
Explanation was provided in line 193 by the addition of:
The 14 days testing time was selected based on the experience from previous studies [Gawel, I.; Pilat, J.; Radziszewski, P.; Niczke, L.; Krol, J.; Sarnowski, M. Bitumen fluxes of vegetable origin. Polimery/Polymers 2010, 55, 55–60, doi:10.14314/polimery.2010.055] as a time when rheological properties stabilize once foamed bitumen recovers its original properties and most of the flux crosslinking process ends.
- on most figures, we see error bars, but we do not know what they represent
Reviewer 2 Report
General comments:
Structure of the article: (Identification of the gap in knowledge): In this study, the authors study the effect of simultaneously using two types of WMA alternatives: fluxing agent (BioFlux), to decrease the production temperature of asphalt mixtures, and water-foaming.
The authors analyze the effect on traditional and performance properties of three types of asphalt binders: paving grade bitumen, polymer-modified bitumen, and highly modified bitumen.
References: Most of the references cited are recent (5 years). The number of self-citations is reasonable.
Relation Hypothesis vs Experimental design: The article doe not clearly declare a hypothesis. Hence, it is not possible to compare the experimental design with the hypothesis.
Reproducibility: The article supplies sufficient information to reproduce the results.
Figures/Tables/Images/Schemes: They properly show the data, and they are easy to interpret and understand.
Conclusions: Conclusions are consistent and Coherent with the analysis and results obtained.
Specific comments:
|
Page 3 Lines 147 |
Add a reference to support that: 50/70 paving grade bitumen, 45/80- 55 polymer modified bitumen and 45/80-80 highly modified bitumen (HiMA), used typically for producing asphalt surface courses intended for low, medium and heavy trafficked roads, respectively.
|
|
Page 4 Line 159 - 185 |
Authors present the characteristics of bio-derived fluxing agent (Bioflux), and the production process. References should be included in order to verify the information.
|
|
Page 5 Line 195
|
Please add information to support why foamed asphalt binders should be tested 14 days after foaming (F-14d)
|
|
Page 5 Line 197. |
Include a description of the preliminary tests used to define the three levels of dose rates.
|
|
Page 5 Line 201 |
Authors indicate that in relevant cases, Pressure Ageing Vessel (PAV) was done to evaluate long-term aging, but results related to this test are not presented in the paper.
|
|
Page 5 – 14 Line 231 - 487 |
The results of all the tests are presented and for each of them an interpretation is shown, but the statistical significant analysis should be included to support the conclusions. Also, statistical analysis will allow identifying data that appears to be inconsistent (As an example Figure 6)
|
|
Pages 488 - 526
|
Conclusions should be supported by statistical analysis.
|
Author Response
Dear Reviewer,
We would like to voice our gratitude for the time and effort spent revising our paper titled “Warm mix asphalt binder utilizing water-foaming and fluxing using bio-derived agent”. We truly feel that your remarks are a significant contribution to the overall quality of our paper and enabled us to rectify its shortcomings.
After a thorough revision, we present you the corrected version of the manuscript for its assessment. Please find the detailed responses to your comments below.
Best regards,
Authors
Reviewer #2:
Structure of the article: (Identification of the gap in knowledge): In this study, the authors study the effect of simultaneously using two types of WMA alternatives: fluxing agent (BioFlux), to decrease the production temperature of asphalt mixtures, and water-foaming.
The authors analyze the effect on traditional and performance properties of three types of asphalt binders: paving grade bitumen, polymer-modified bitumen, and highly modified bitumen.
References: Most of the references cited are recent (5 years). The number of self-citations is reasonable.
Relation Hypothesis vs Experimental design: The article doe not clearly declare a hypothesis. Hence, it is not possible to compare the experimental design with the hypothesis.
Reproducibility: The article supplies sufficient information to reproduce the results.
Figures/Tables/Images/Schemes: They properly show the data, and they are easy to interpret and understand.
Conclusions: Conclusions are consistent and Coherent with the analysis and results obtained.
Thank you for the generous comments.
Specific comments:
|
Page 3 Lines 147 |
Add a reference to support that: 50/70 paving grade bitumen, 45/80- 55 polymer modified bitumen and 45/80-80 highly modified bitumen (HiMA), used typically for producing asphalt surface courses intended for low, medium and heavy trafficked roads, respectively.
Thank you for the remark. We have corrected the sentence and provided an adequate reference in line 145: “(…) used in Poland [44] typically for producing asphalt surface courses intended for low, medium and heavy trafficked roads, respectively”
|
|
Page 4 Line 159 - 185 |
Authors present the characteristics of bio-derived fluxing agent (Bioflux), and the production process. References should be included in order to verify the information.
Thank you for the remark. Authors provided an additional information by the following addition to the manuscript: “based on the experience from previous studies [25, 29]”, following previously existing text “The fluxing agent was produced using pure fatty acid methyl esters derived from rapeseed oil (RME) without adding anti-ageing additives”
|
|
Page 5 Line 195
|
Please add information to support why foamed asphalt binders should be tested 14 days after foaming (F-14d) The 14 days testing time was selected based on the experience from previous studies [Gawel, I.; Pilat, J.; Radziszewski, P.; Niczke, L.; Krol, J.; Sarnowski, M. Bitumen fluxes of vegetable origin. Polimery/Polymers 2010, 55, 55–60, doi:10.14314/polimery.2010.055] as a time when rheological properties stabilize once foamed bitumen recovers its original properties and most of the flux crosslinking process ends. Explanation was provided in line 193 by the addition of: The 14 days testing time was selected based on the experience from previous studies [Gawel, I.; Pilat, J.; Radziszewski, P.; Niczke, L.; Krol, J.; Sarnowski, M. Bitumen fluxes of vegetable origin. Polimery/Polymers 2010, 55, 55–60, doi:10.14314/polimery.2010.055] as a time when rheological properties stabilize once foamed bitumen recovers its original properties and most of the flux crosslinking process ends. |
|
Page 5 Line 197. |
Include a description of the preliminary tests used to define the three levels of dose rates. An additional explanation was provided in line 197 “time-related stabilization and experiences from previous studies [25, 29, 30]”
|
|
Page 5 Line 201 |
Authors indicate that in relevant cases, Pressure Ageing Vessel (PAV) was done to evaluate long-term aging, but results related to this test are not presented in the paper.
Thank you for your remark. We have supplemented section 2.2.2 with the information that the presented MSCR results were obtained on RTFOT aged samples and the BBR samples were subjected to RTFOT+PAV ageing
|
|
Page 5 – 14 Line 231 - 487 |
The results of all the tests are presented and for each of them an interpretation is shown, but the statistically significant analysis should be included to support the conclusions. Also, statistical analysis will allow identifying data that appears to be inconsistent (As an example Figure 6)
Thank you for the remark. We have added statistical analyses and adequate descriptions to the results.
|
|
Pages 488 - 526
|
Conclusions should be supported by statistical analysis.
Thank you for the remark. We have added statistical analyses and adequate descriptions to the results. |
Reviewer 3 Report
This research work aims to analyze the effects of simultaneous water-foaming and fluxing on classical and performance related properties of three asphalt binders (paving grade bitumen, polymer modified bitumen and a highly modified bitumen). It is well written and presents relevant results considering the use of bio-derived agent. Congratulations. Some suggestions are presented below:
11) Please add a discussion about this topic here: “Additionally, the dynamic viscosities of the asphalt binders.”.
22) For the section 2.2.2., please, add a schematic representation of the work in order to facilitate the visualization of the methods and the objectives.
33) Any information about LAS test for the binders? And what about PAV?
44) Why were the results of penetration similar for all cases of the binder 45/80-55 (NF, F, and F-14-d)?
55) Please, explain better this sentence (line 245-247): “The non-foamed asphalt binders amounted to, respectively, about 25·0.1 mm for 50/70 and 45/80-55 binders and approx. 18·0.1 mm for the 45/80-80 highly modified binder”. Is this related to the increase of bioflux from 0 to 2%?
66) Any explanation for the samples presented an increase in the change in dynamic viscosity, mainly 50/70 F-14d 3% of fluxing agent?
77) I would suggest comparing all results with similar works from the literature.
88) Also, the authors can omit the graph of the binder 50/70 of the Figure 9.
99) Avoid using references in conclusion section.
Author Response
Dear Reviewer,
We would like to voice our gratitude for the time and effort spent revising our paper titled “Warm mix asphalt binder utilizing water-foaming and fluxing using bio-derived agent”. We truly feel that your remarks are a significant contribution to the overall quality of our paper and enabled us to rectify its shortcomings.
After a thorough revision, we present you the corrected version of the manuscript for its assessment. Please find the detailed responses to your comments below.
Best regards,
Authors
Reviewer #3:
This research work aims to analyze the effects of simultaneous water-foaming and fluxing on classical and performance related properties of three asphalt binders (paving grade bitumen, polymer modified bitumen and a highly modified bitumen). It is well written and presents relevant results considering the use of bio-derived agent. Congratulations. Some suggestions are presented below:
Thank you for the generous comments.
11) Please add a discussion about this topic here: “Additionally, the dynamic viscosities of the asphalt binders.”.
Thank you for the comment. We have corrected the sentence:
“Additionally, the dynamic viscosities of the asphalt binders were significantly affected by the modification.“
22) For the section 2.2.2., please, add a schematic representation of the work in order to facilitate the visualization of the methods and the objectives.
Thank you for the remark. We have improved the section 2.2.2 by describing the investigated factors and clarifying the ageing protocols relevant to particular tests.
33) Any information about LAS test for the binders? And what about PAV?
Thank you for the remark. LAS testing was not performed in this study. We have supplemented section 2.2.2 with the information that the presented MSCR results were obtained on RTFOT aged samples and the BBR samples were subjected to RTFOT+PAV ageing.
44) Why were the results of penetration similar for all cases of the binder 45/80-55 (NF, F, and F-14-d)?
Thank you for the remark. We have supplemented the paper with the statistical analysis which has verified the statements provided in the manuscript. We have also provided a reference to a study including investigation of foamed asphalt binders in a similar scope conducted by Martinez-Arguelles et al. (Martinez-Arguelles, G.; Giustozzi, F.; Crispino, M.; Flintsch, G.W. Investigating physical and rheological properties of foamed bitumen. Constr. Build. Mater. 2014, 72, 423–433, doi:10.1016/j.conbuildmat.2014.09.024.) It is often found that depending on the type of the binder, it responds differently to the foaming process – the responses in scope of classical characteristics of asphalt binders are observed to differ in different binders, just as seen in our study.
55) Please, explain better this sentence (line 245-247): “The non-foamed asphalt binders amounted to, respectively, about 25·0.1 mm for 50/70 and 45/80-55 binders and approx. 18·0.1 mm for the 45/80-80 highly modified binder”. Is this related to the increase of bioflux from 0 to 2%?
Thank you for the remark. We have clarified the sentence:
„The change in penetration caused by evey percent of Bioflux content in non-foamed asphalt binders amounted to, respectively, about 25·0.1 mm for 50/70 and 45/80-55 binders and approx. 18·0.1 mm for the 45/80-80 highly modified binder.”
66) Any explanation for the samples presented an increase in the change in dynamic viscosity, mainly 50/70 F-14d 3% of fluxing agent?
Thank you very much for taking our attention on the outliers. Authors reinvestigated Figure 6 and discovered editorial mistake (binder 50/70, F14ds). Improved figure was replaced with the previous one.
77) I would suggest comparing all results with similar works from the literature.
Thank you for the remark. We have provided comparisons with relevant studies where it was possible (classical characteristics of asphalt binders, Section 3.1.1), however, there are not many relevant studies including simultaneously foamed and fluxed asphalt binders.
88) Also, the authors can omit the graph of the binder 50/70 of the Figure 9.
Thank you for the suggestion. We have omitted the 50/70 binder in Figure 9 and adjusted the text accordingly.
99) Avoid using references in conclusion section.
Thank you for the remark.
Reviewer 4 Report
In this contribution, the authors investigated the effects of foaming and a fluxing agent on the performance of three asphalt binders. The results showed that the foaming had only minor and short-term impacts, but the fluxing agent caused significant changes in binders. This research is inspiring to the readership of Materials. Therefore, I would look forward to replies to the following questions and comments before recommend its publishing.
1. The oxidation and cross-linking efficiency of RME was tested in the presence of a cobalt catalyst and the cumene hydrogen peroxide initiator. Is the temperature of 25 °C sufficient to generate free radicals from the initiator? Is the oxygen from the airflow a scavenger of radicals? What are the results of oxidation and cross-linking efficiency of RME?
2. Why do foamed binders show decreased penetration? If the foaming increases the free volume in binders, should the penetration increase?
3. For the bonders 50/70 and 45/80-55, the softening points lowered by 1 °C after foaming, and the difference disappeared after 14 days. Are the lower softening points, though subtle, due to the foaming-induced free volume? Does the free volume disappear after 14 days due to the relaxation of binders? Contrarily, does 45/80-80 retain the free volume because of its higher polymer content? Can the evolution of free volume be characterized by microscopy?
4. How good is the compatibility between the RME and binders, especially 45/80-80? When the testing temperature is higher than the softening point, the results show fluctuations, e.g., 45/80-8F at 110/135 °C in Figure 6. Is the fluctuation due to the phase separation between RME and binders?
5. Other amendments.
Line 282 states the 45/80-80 binder being the exception of foaming increasing the breaking point. Is 45/80-80 or 45/80-55 the exception?
Line 111, 300-400 °C.
The overlapping curves in Figure 5 are illegible.
Author Response
Dear Reviewer,
We would like to voice our gratitude for the time and effort spent revising our paper titled “Warm mix asphalt binder utilizing water-foaming and fluxing using bio-derived agent”. We truly feel that your remarks are a significant contribution to the overall quality of our paper and enabled us to rectify its shortcomings.
After a thorough revision, we present you the corrected version of the manuscript for its assessment. Please find the detailed responses to your comments below.
Best regards,
Authors
Reviewer #4:
In this contribution, the authors investigated the effects of foaming and a fluxing agent on the performance of three asphalt binders. The results showed that the foaming had only minor and short-term impacts, but the fluxing agent caused significant changes in binders. This research is inspiring to the readership of Materials. Therefore, I would look forward to replies to the following questions and comments before recommend its publishing.
Thank you for the generous remarks.
- The oxidation and cross-linking efficiency of RME was tested in the presence of a cobalt catalyst and the cumene hydrogen peroxide initiator. Is the temperature of 25 °C sufficient to generate free radicals from the initiator? Is the oxygen from the airflow a scavenger of radicals? What are the results of oxidation and cross-linking efficiency of RME?
We appreciate the reviewer for this question.
Shorter time and lower temperatures (20-30 °C) of oxidation of RMSs promote the formation of products with higher hardening potential due to the preservation of more reactive double bonds and a higher content of peroxide and hydroperoxide structures. It is a well-known phenomenon that using peroxide polymerization initiators during the oxidation of vegetable oils or oil esters at a lower temperature increases the hardening effect of liquefied binders. For oxidation, a special reactor was used to obtain a higher increase in the peroxide number while maintaining a high iodine value of the oxidized raw material in the presence of a cobalt catalyst, an active oxidation catalyst. A promoter of the oxidative polymerization reaction was also used, which would allow the process to be carried out at ambient temperature. As initiators of the polymerization reaction, cumene hydrogen peroxide was used, which is widely used in curing unsaturated resins, e.g. polyester. The presence of a cobalt catalyst causes a reduction of peroxides introduced into alkoxyl radicals at a temperature of 25°C, as a result of which the polymerization reaction takes place at temperatures lower than the half-life temperature of the promoter. Therefore, the catalytic oxidation of plant raw materials in the presence of peroxides was carried out at an ambient temperature of 25°C.
- Why do foamed binders show decreased penetration? If the foaming increases the free volume in binders, should the penetration increase?
Thank you for the comment. See our joint response below.
- For the bonders 50/70 and 45/80-55, the softening points lowered by 1 °C after foaming, and the difference disappeared after 14 days. Are the lower softening points, though subtle, due to the foaming-induced free volume? Does the free volume disappear after 14 days due to the relaxation of binders? Contrarily, does 45/80-80 retain the free volume because of its higher polymer content? Can the evolution of free volume be characterized by microscopy?
Thank you for the accurate comments. We have supplemented the paper with the statistical analysis which has verified the statements provided in the manuscript. We have also provided a reference to a study including investigation of foamed asphalt binders in a similar scope conducted by Martinez-Arguelles et al. (Martinez-Arguelles, G.; Giustozzi, F.; Crispino, M.; Flintsch, G.W. Investigating physical and rheological properties of foamed bitumen. Constr. Build. Mater. 2014, 72, 423–433, doi:10.1016/j.conbuildmat.2014.09.024.) It is often found that depending on the type of the binder, it responds differently to the foaming process – the responses in scope of classical characteristics of asphalt binders are observed to differ in different binders, just as seen in our study. What is significant is the fact, that these effects dissipate over time, as the moisture acquired during foaming dissipates in the bitumen matrix. Huang et al. (B. Huang, Y. Zhang, X. Shu, Y. Liu, D. Penumadu, X.P. Ye, Neutron Scattering for Moisture Detection in Foamed Asphalt, J. Mater. Civ. Eng. 25 (2013) 932–938,) has shown that mixroscopic water droplets are not detectable in foamed bitumen, while Hung et al. (A.M. Hung, A. Goodwin, E.H. Fini, Effects of water exposure on bitumen surface microstructure, Constr. Build. Mater. 135 (2017) 682–688, https://doi.org/10.1016/j.conbuildmat.2017.01.002) has shown that water may permeate into and interact with the bitumen. The results of these studies are however very loosely tied to the obtained results and therefore were not cited nor discussed.
The free volume gained due to foaming probably decreases rapidly in room temperature to the mentioned phenomenon of relaxation, however we have not performed such analyses.
- How good is the compatibility between the RME and binders, especially 45/80-80? When the testing temperature is higher than the softening point, the results show fluctuations, e.g., 45/80-8F at 110/135 °C in Figure 6. Is the fluctuation due to the phase separation between RME and binders?
We appreciate the reviewer for this question.
It should be stated that RME and bitumen show full miscibility and are compatible. No separation was observed between RME and binder phases. During the modification, the mixing process was carried out to complete the homogeneity of the mixture. The fluctuation that occurred is most likely due to the very high viscosity of binder 45/80-80, which may increase the viscosity measurement error, especially observed in PMB studies.
- Other amendments.
Line 282 states the 45/80-80 binder being the exception of foaming increasing the breaking point. Is 45/80-80 or 45/80-55 the exception?
Thank you for the remark. We have clarified the sentence:
“In cases of the highly modified 45/80-80 binder, the softening point was reduced to significantly higher degree immediately after and 14 days after foaming when dosed with 1% and 3% of Bioflux agent.”
Line 111, 300-400 °C.
Thank you for the remark.
The overlapping curves in Figure 5 are illegible.
Thank you for the remark. We have attempted to increase the legibility of the figure by increasing its size, linewidths and adding some transparency to the lines.
Round 2
Reviewer 1 Report
Paper is much better now. The results are better analysed. good job